# 500-kHz Level High Energy Double-Pass Nd:YVO$_4$ Picosecond Amplifier with Optic–Optic Efficiency of 51%

**Xuesheng Liu** [1,2,3,*], **Huan He** [1,2,3], **Yiheng Song** [1,2,3,4], **Congcong Wang** [1,2,3] and **Zhiyong Wang** [1,2,3]

[1] Institute of Laser Engineering, Beijing University of Technology, Beijing 100124, China; hehuan2016@emails.bjut.edu.com (H.H.); 33nature@126.com (Y.S.); wccong@emails.bjut.edu.cn (C.W.); zywang@bjut.edu.cn (Z.W.)

[2] Beijing Engineering Research Center of Laser Technology, Beijing 100124, China

[3] Key Laboratory of Trans-scale Laser Manufacturing Technology, China Ministry of Education, Beijing 100124, China

[4] The 53rd Research Institute of the China Electronics Technology Group Corporation, Tianjin 300000, China

[*] Correspondence: liuxuesheng@bjut.edu.cn; Tel.: +86-138-1043-5431

**Abstract:** We have demonstrated a high pulse energy and high optic–optic efficiency double-pass picosecond (ps) master oscillator power amplifier system of 1064 nm at a pulse repetition rate of 500 kHz. A 500 kHz, 7.68 µJ picosecond laser is used as the seed laser. Through one stage double-pass traveling-wave amplifier, a maximum output power of 16.19 W at a pump power of 31.7 W is generated with the optic–optic efficiency of 51.07%. The output pulse duration is 17.6 ps, corresponding to the pulse energy of 32.38 µJ. The beam quality factor $M^2$ were measured to be 1.28 and 1.17 along the x, y axis direction, respectively.

**Keywords:** double-pass amplifier; high energy; high optic–optic efficiency; 500 kHz

## 1. Introduction

Due to the obvious advantages in research and industrial fields [1–4] in recent years, much work has been dedicated to high-power, high-repetition-rate picosecond lasers, especially those with good beam quality, narrow pulse width. The most popular way of achieving the above-mentioned laser characteristics is the "MOPA" (master oscillator power amplifier) approach [5–8]. Compared with the Q-switched method, the direct-modulated semiconductor diode is more suitable as the master oscillator, because of its abilities of high pulse repetition frequency (PRF).

Nd:YVO$_4$ crystals as a gain material with sufficient gain bandwidth and a high emission cross section are widely adopted in end-pumping picosecond regenerative amplifiers. It has larger stimulated emission of radiation at 1.06 µm, which is about five times as large as that of Nd:YAG. The absorption bandwidth around 808 nm is about five times as much as that of Nd:YAG, and the absorption coefficient is so large that it is conductive for LD (Laser Diode) pumped [9,10]. Existing experiments show that Nd:YVO$_4$ is a kind of laser crystal with low threshold and high efficiency. While it is also a type of crystal with high birefringence ($n_o$ = 1.958, $n_e$ = 2.168 at 1.06 µm), which is conducive for avoiding the thermal birefringence of Nd:YAG.

In 2016, Shen Lifeng [11] reported a picosecond laser amplifier based on grazing incidence Nd:YVO$_4$ slab geometry. The microchip laser was used as seed laser with pulse duration of 90 ps, repetition rate of 100 kHz and beam quality factor $M^2$ = 1.16. An output power of 13 W was obtained with the liquid pure metal grease conducted as the thermal contact material for 10 mW seed power

with pump power of 55 W at an optical–optical efficiency of 23%, achieving pulse peak power of 1.2 MW and pulse energy of 130 μJ. The beam quality factor was $M_x^2 = 1.30$, $M_y^2 = 1.28$.

In 2016, Ying Chen [12] had demonstrated a high pulse energy, high peak power, and high beam quality picosecond (ps) master oscillator power amplifier system of 1064 nm at a pulse repetition rate of 5 kHz. This system consisted of a ps mode-locked oscillator, a regeneration amplifier, and three-stage Nd:YAG slab amplifiers, which enabled us to manage good thermal issues and provide an average power of 41 W. Up to 8.2 mJ of energy per pulse was achieved at the total absorbed pump power of 460 W with an average extraction efficiency of 8.5%. The pulse duration was 25.3 ps, which corresponds to a peak power of 324 MW. The beam quality factors of $M^2$ were measured to be $M_x^2 = 2.8$ along the x-axis direction and $M_y^2 = 2.2$. along the y-axis direction, respectively.

In 2017, Zhenao Bai [13] reported a non-pulse-leakage optical fiber pumped 100 kHz level high beam quality Nd:YVO$_4$ picosecond amplifier. An 80 MHz, 11.5 ps mode-locked picosecond laser was used as the seed with single pulse energy of 1 nJ, a maximum output power of 24.5 W was generated corresponding to the injected regenerative amplified power of 9.73 W at 500 kHz. The output pulse duration is 16.9 ps, and the beam quality factor $M^2$ is 1.25, the optic–optic efficiency of amplifier is 39.7%.

In 2018, Wang Yong [14] reported a high peak power and high beam quality fiber—a solid hybrid amplification laser system. A fiber laser with a repetition frequency of 50 kHz, pulse width of 3.9 ps, and average power of 10.9 mW was used as the seed source. The average output power of 27.65 W was obtained in the double-passing amplifier of two stages, with the pulse peak power of 65 MW. The first amplifier stage is end-pumped Nd:YVO$_4$ solid amplifier and the second is side-pumped Nd:YAG solid amplifier. By optimizing the beam filling factor in each solid amplifier, the $M^2$ factor of the output beam reaches 1.30.

In this study, we demonstrate a highly stable and high beam quality industrial grade double-pass picosecond Nd:YVO4 amplifier with repetition rate of 500 kHz. The average output power of 16.19 W was obtained at 500 kHz with the pulse width of 17.6 ps and a pump power of 31.7 W, corresponding to the output pulse energy of 32.38 μJ. The beam quality factors of $M^2$ were measured to be $M_x^2 = 1.28$ along the x axis direction and $M_y^2 = 1.17$ along the y axis direction, respectively. Compared with the results of existing amplifiers [11–13], the optic–optic efficiency of 51.07% is higher with good beam quality and high frequency. Our measurements show that this amplifier structure enables the laser to be fully amplified and operates stably. The amplifier can be used in the fine processing of hard and brittle materials, such as laser drilling, laser scribing, laser cutting for glass, ceramics, silicon wafers, etc.

## 2. Theoretical Analysis

The energy required for laser amplification comes from the energy stored in the gain medium before signal input. For lasers, the expression of laser pulse amplification gain [15] is

$$G = \frac{E_s}{E_0} ln \left\{ 1 + \left[ \exp\left( \frac{mE_0}{E_s} \right) - 1 \right] - G_0 \right\} \tag{1}$$

where $E_s$ is the saturation energy density; $E_0$ is the energy density of the input pulse; $G_0$ is small signal gain coefficient.

The relation between small signal gain and pump energy is

$$G_0 = \exp(g_0 l) = \exp(\beta E_s l) \tag{2}$$

where $g_0$ is the gain coefficient of unit length in the gain medium, $l$ is the length of the gain medium;
And in a four-level system

$$E_s = \frac{h\nu}{\sigma} \tag{3}$$

where $h$ is the Planck constant; $\nu$ is the laser frequency; $\sigma$ is Excited interface. Therefore, the output energy of the multistage laser amplification system is

$$E = (\frac{E_s}{m})ln\left\{1 + \left[\exp\left(\frac{mE_0}{E_s}\right) - 1\right]\exp(m\sigma N_0)\right\} \quad (4)$$

$$N_0 = \frac{g_0 l}{\sigma} \quad (5)$$

where $m$ is the number of times the laser passes through the gain medium, and $g_0 l$ is the one-way small signal gain factor.

In our experiments, $h = 6.6260755 \times 10^{-34}$ J·s, $\nu = 500$ kHz, $\sigma = 25 \times 10^{-19}$ cm$^{-2}$, $E_s = 1.3252$ μJ/m$^2$, $E_0 = 24.45$ μJ/m$^2$, $m = 2$, $l = 20$ mm. After theoretical calculation, theoretical output pulse energy is 26.78 μJ was obtained.

## 3. Experimental Setup

In the experiment, a MOPA system that consisted of a ps mode-locked oscillator, a four-pass traveling-wave amplifier, and a double-pass traveling-wave Nd:YVO$_4$ amplifier was adopted to obtain the high pulse energy output at 1064 nm, presented in Figure 1.

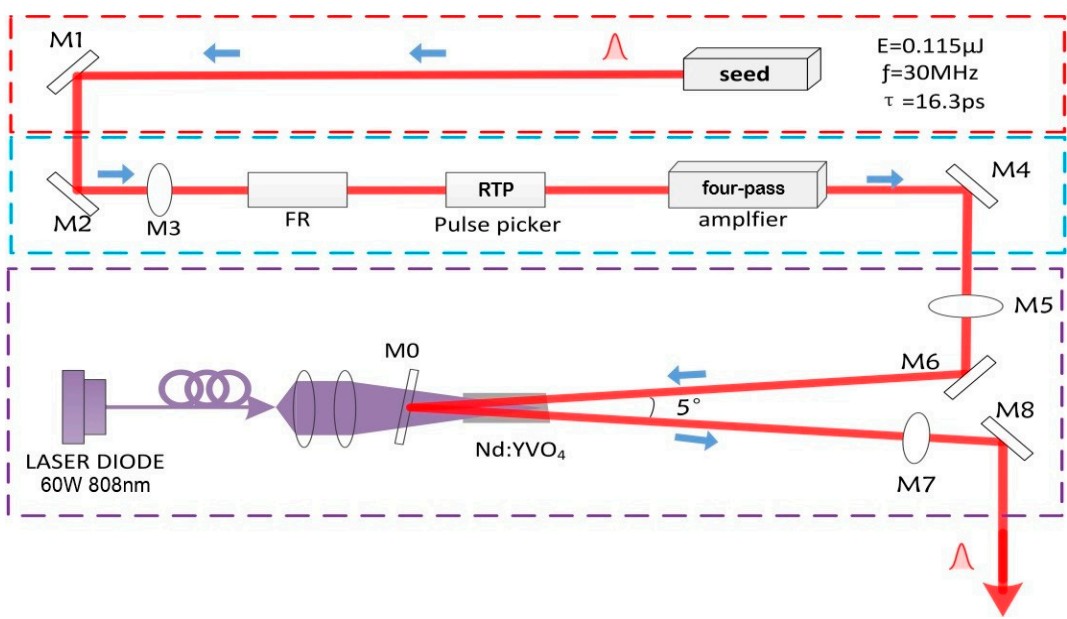

**Figure 1.** Schematic diagram of the 500-kHz level high energy double-pass Nd:YVO$_4$ picosecond amplifier experimental setup.

In the present system, the amplifier is seeded with a self-developed semiconductor saturable absorption mirror (SESAM) mode-locked Nd:YVO$_4$ oscillator capable of generating 16.3 ps pulses at a repetition rate of 30 MHz with a single pulse energy [16] of 0.115 μJ. Picosecond pulse sequence from the seed was injected into the pulse picker through two high-reflection mirrors, M1 and M2. The function of the pulse picker is to select the desired repetition rate for traveling amplifier by eliminating the pulse leakage of the seed sequence. The size of the RTP (Rubidium Titanyl Phosphate) crystals used in the pulse picker is $5 \times 5 \times 7$ mm$^3$.

The laser pulse is injected into the four-pass amplifier by the M3. In the four-pass amplifier, the gain medium is 0.3 at.% doped Nd:YVO$_4$ with a size of $4 \times 4 \times 12$ mm$^3$ and double-end-wedged cut at 2°. Both sides of the Nd:YVO$_4$ have anti-reflective (AR) coating at the 1064 nm and 888 nm wavelength. A 60 W, 888 nm fiber-coupled laser diode (p2-060-0888-3-A-R01-S0119 made by Nlight) with a numerical aperture of 0.22 and a diameter of 400 μm is used as the pump source. The coupling ratio of the pump beam is 1:2 into the Nd:YVO$_4$ crystal.

To further amplify the output power, one stage traveling-wave Nd:YVO$_4$ amplifier is adopted. Through a focus lens M5 and two high-reflection mirrors, M4, M6, the pulses were injected into the

double-pass traveling-wave amplifier. A Nd:YVO$_4$ crystal with the dimensions of $4 \times 4 \times 20$ mm$^3$ is selected as gain medium with 0.3 at.% Nd$^{3+}$-doped and double-end-wedged cut at 2° to inhibit the production of ASE. Both end facets of the laser crystal were anti-reflection (AR) coated at 1064 nm and 808 nm. A 60 W, 808 nm fiber-coupled laser diode (SN: 1038266-W4312 made by Nlight) with the numerical aperture of 0.22 and the diameter of 400 μm was used as the pump source of the amplifier, the line width is ±3 nm. Reflected by M0, the laser passes the gain medium two times with the coupling ratio of 1:4. The folding angle of the light path was about 5°.

In the system, The plane mirrors M1, M2, M4, M6 and M8 were HR coated at 1064 nm and 808 nm; M3 was a focus lens with anti-reflective (AR) coating at the 1064 nm wavelength of size φ12.7 × 6 (f = 200 mm), M5 and M7 were focus lens with antireflective (AR) coating at the 1064 nm and 808 nm wavelength of size φ12.7 × 4 (f = 150 mm).The arm length between M4~M5 and M5~M6 was carefully optimized to be 75 mm and 45 mm; the arm length between M0 and M6 was 265 mm; the arm length between M0 and M8 was 327 mm. The offset in x, y axis is 0.3 mm and the fabrication tolerance is ±0.01 mm. In order to reduce the thermal effects and improve laser beam quality at high power, the temperature of the water was maintained at 19 °C.

## 4. Experimental Results and Discussion

In our experiment, the pump module, collimation system, coupling-out mirror and Nd:YVO$_4$ crystal were adjusted to be coaxial. The LD current was adjusted to obtain output power with the wavelength 1064 nm, at the maximum pump power of 18.66 W, the output power of the four-pass amplifier was 3.84 W at 500 kHz. After passing through the double-pass amplifier, a maximum amplified power of 16.19 W was obtained corresponding to the pulse energy of 32.38 μJ at a pumped current of 4 A. As shown in Figure 2. At a pump current of 1 A, the output power was 1.73 W; at a pump current of 7.4 A, a max pump power of 60 W and a output power of 26.4 W were obtained. The pump power of the double-pass amplifier was set to be 31.7 W because higher power may result in the self-excited oscillation. It can be observed that the output power increases linearly with the increasing of the pumped power and there are two turning points in the image. This is because the crystal has a strong absorption of 808 nm pump light, the efficiency will be greater when the current is less than 2 A, but the thermal effect gradually increases with the pump current gradually increasing, and the efficiency will gradually decrease, and it will decrease further when it reaches 5 A, therefore there are two turning points in the curve.

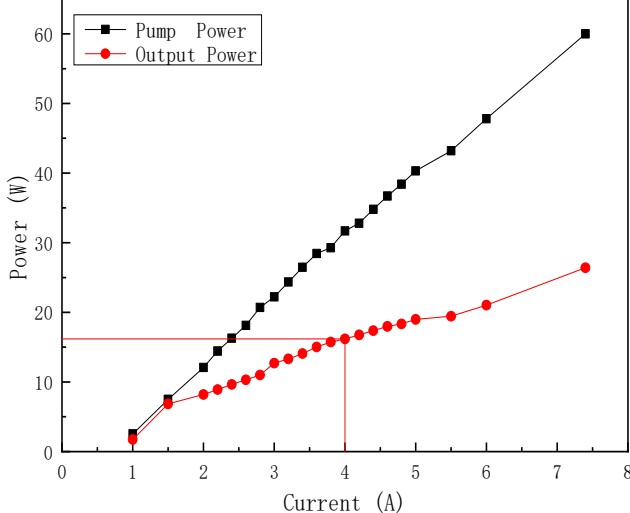

**Figure 2.** Output power of 808 nm LD and the 1064 nm output laser versus pump current.

The amplified pulses were monitored using a photodiode (ALPHALAS GMBH, UPD-40-UVIR-D, rise time < 40 ps). At 500 kHz, oscilloscope trace of the output pulse train is illustrated in Figure 3. We can observe that the amplifier generates very clean output pulses.

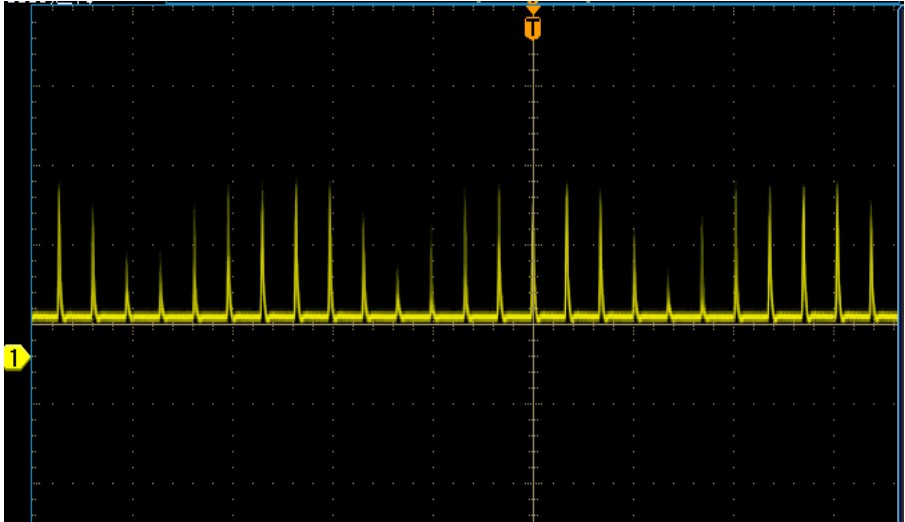

**Figure 3.** Oscilloscope trace of the double-pass amplified pulse train.

Figure 4a shows the autocorrelation trace of the amplified pulses with the measured width of 17.6 ps at a repetition rate of 500 kHz. The pulse width of the output was slightly broadened compared with that of the seed pulses 16.3 ps. At output energy of 32.38 μJ, we measured the laser beam profile of the focused output beam using the knife-edge method [17], as is shown in Figure 4b. The $M^2$ measured were 1.28 and 1.17 for the horizontal and vertical axes of the output, respectively. The roundness of the near-field intensity distribution is higher than 99%.

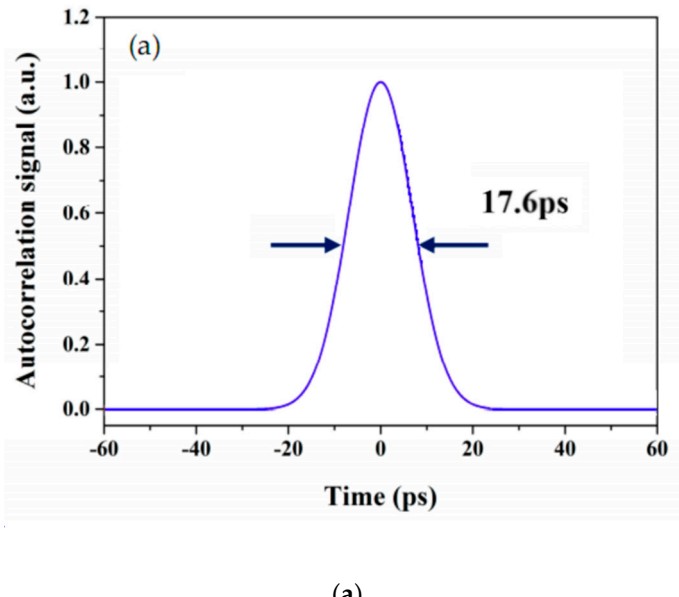

(**a**)

**Figure 4.** *Cont*.

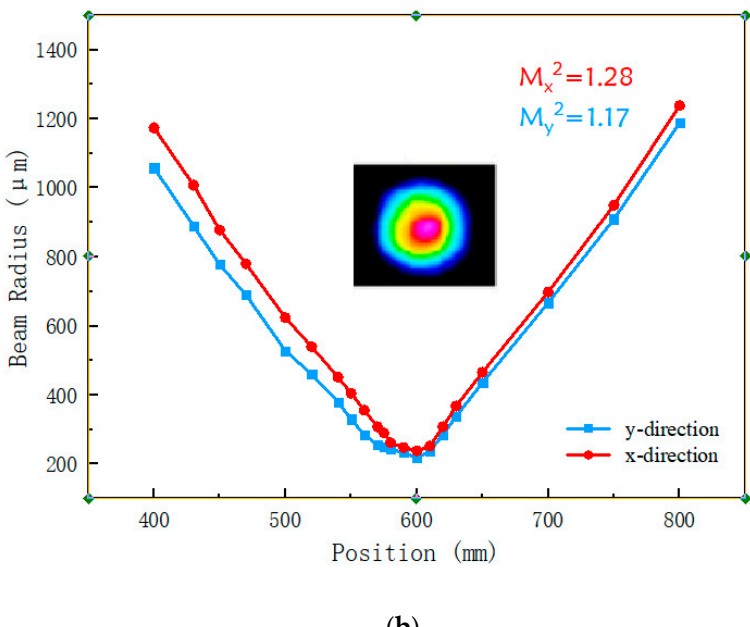

**(b)**

**Figure 4.** (**a**) Measured autocorrelation trace for the amplified pulse; (**b**) measured beam radius and fitted beam quality for the tangential direction ($\omega_x$) and the sagittal direction ($\omega_y$).

It could be observed that there were differences between beam quality in the x direction and y direction. The main cause was the asymmetry of crystal cooling structure. $Nd:YVO_4$ crystals are sensitive to changes of temperature, so by optimizing the cooling structure of crystals in different directions, the beam quality in the x direction can be optimized. Compared to the reported papers about picosecond laser amplifier, the optic–optic conversion efficiency 51.07% obtained by this paper was the highest, and the beam quality $M^2$ was also better. Experiments show that the laser has good stability; firstly, we cut the crystal with an angle to prevent the generation of ASE; In addition, the focus mirrors and the end surface of $Nd:YVO_4$ are AR coated with transmittance greater than 99.5%, and the crystal has a large static distance from the focus mirror, so there will be no resonance in the amplifier, the effect of back reflection is minimal in the laser and the laser runs well.

## 5. Conclusions

In conclusion, we had demonstrated a 500 kHz level high energy double-pass $Nd:YVO_4$ picosecond amplifier with optic–optic efficiency of 51.07%. This design successfully amplified the seed beam and ensured the beam quality. An average power of 16.19 W was obtained at 500 kHz, and a maximum single pulse energy of 32.38 µJ was obtained at a pumped power of 31.7 W. High beam quality output was obtained with $M^2$ about 1.28 and 1.17 for the horizontal and vertical axes. The output pulse width is 17.6 ps. Our work had controlled the thermal effect while passing the gain medium multitimes and ensured the beam quality. Furthermore, this picosecond amplifier could be a promising candidate for improving optical-optical efficiency [18].

**Author Contributions:** X.L. conceived and designed the experiments; H.H. and C.W. performed the experiments; Y.S. analyzed the data; Z.W. contributed reagents/materials/analysis tools; H.H. wrote the paper.

**Funding:** This work was supported by a grant from the National Key R&D Program of China (Grant No. 2017YFB0305800).

**Acknowledgments:** The authors acknowledge Mr. Yan for his help in the experiment.

**Conflicts of Interest:** The authors declare no conflict of interest.

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
