# Peer review of "500-kHz Level High Energy Double-Pass Nd:YVO4 Picosecond Amplifier with Optic–Optic Efficiency of 51%"

_applsci, doi:10.3390/app9020219_

Round 1

Reviewer 1 Report

In this paper, the authors introduced a high pulse energy and high optic-optic efficiency double-pass 13 picosecond (ps) master oscillator power amplifier system of 1064 nm at a pulse repetition rate of 14 500 kHz. A 500 kHz, 7.68 μJ picosecond laser is used as the seed laser. Through one stage double15 pass traveling-wave amplifier, a maximum output power of 16.19 W at a pump power of 31.7 W is generated with the optic-optic efficiency of 51.07%.

The idea behind this is interesting. However, I still have quite a number of concerns in this manuscript. There are times where there are not enough data to support the conclusions of the author. Please see some of the major concerns below.

1.The information of the optical setup is not enough (figure 1). The authors should give much more information about this. So the readers can get its reproducibility. 

2.  The authors should give much more information about the novelty of this paper, especially the effect of using this high pulse energy which applications can used it?

3. The fabrication tolerance analysis, which can offer a good guide for the fabrication requirement, and the key parameters (BR, offset in x,y,x axis), need to be added in the results section.

4. More references need to be included in the introduction part to understand the applications of using high pulse energy and laser applications:

       1."Prospects for diode-pumped alkali-atom-based hollow-core photonic-crystal fiber    lasers",Optics Letters, 39(16), 2014 (4655-4658)

     .  2."A robust all-fiber Q-switched 1 -mm Yb3+ fiber laser"

       Applied Physics B: Lasers and Optics, ;120(3), 2015 (489-495)

      .3. "Design of 4 x 1 power beam combined based on multicore photonic crystal fiber"

       Applied Sciences, 7(7), 2017 (695 – 9 pages).

5. Authors needs to analysis the back reflection coming to the laser direction.

Author Response

1.       The information of the optical setup was added in FIG1.

2.       More information about the novelty was added in line 69-73

      3.  The key parameters were added in line 124-125

      4.  More references were added in line 26.

      5.  The explanation was added in line168 and line114

Reviewer 2 Report

1. Authors have mentioned in the title “optic-optic efficiency of 51%“. Is it “optic-optic” or “optical-optical” ?

2. In page 2; line 68: authors have written the following sentence:

“Compared with the results of previous works, the optic-optic efficiency of ……”.

Author should provide references for the previous works.

3. In the experimental set up section, authors have mentioned that a 60 W, 888 nm fiber-coupled laser diode is used as the pump source. However, they have mentioned 808 nm LD in caption of Figure 2. What is the linewidth of the laser diode ?

4. Why does the 1064 nm laser output power show nonlinear behavior ? Authors should explain the observed results.

5. Authors should explain the variation of 1064 nm laser output power with respect to optical pump power.

6. Authors should provide scientific explanation to their experimental observations.  

Author Response

1. It was “optic-optic”. 2.  In page 2; line 68, “Previous works” was corrected and references was added. 3.  The 60 W, 888 nm fiber-coupled laser diode is used as the pump source was used in the 4-pass amplifier and 808nm LD was used in the double-pass amplifier. The linewidth was added in line 118. 4.  The explanation was given in the line 137-142. 5.  The explanation was given in the line137-142. 6.  The explanation was given in the article.

Round 2

Reviewer 1 Report

 The new version much better so now the paper can be published

Reviewer 2 Report

I appreciate authors' efforts.